# OpenReview forum: "LLM Censorship: The Problem and its Limitations"
_ICLR.cc/2024/Conference — Submitted to ICLR 2024_

### Official Review · Reviewer_BZEx · 2023-10-24

**Soundness:** 2 fair
**Presentation:** 4 excellent
**Contribution:** 2 fair
**Rating:** 5
**Confidence:** 4

**Summary:**

The paper focuses on the problem of "censorship" in large language models (LLM). Specifically, the paper argues that it is unfeasible to address this issue by relying on ancillary "machine learning" (ML) techniques, and that it should rather be tackled via mechanisms belonging to the security domain. To support such a position, the paper presents detailed theoretical arguments demonstrating that LLM censorship is an "undecidable problem", thereby revealing that using ML-based techniques, such as, e.g., another language model (LM), will never provide a foolproof solution.

**Strengths:**

## High-level

+ Outstanding writing
+ Relevant Problem (for both research and practice)
+ The theoretical arguments are well-founded

## Comment

I deeply thank the authors for writing this piece and submitting it to ICLR'24. I've loved reading it, and I was genuinely pleased by the outstanding writing quality: out of the papers I reviewed for ICLR'24, this one is by far the best written one. Moreover, the paper tackles a very open issue and the "conclusion" can be leveraged by researchers and practitioners alike: the latter can benefit by integrating additional security mechanisms in their products, whereas the former would be provided with "clear evidence" that tackling censorship by means of traditional ML methods will never provide a foolproof solution. Indeed, the theoretical arguments made in this paper are well-rooted, and I particularly appreciated connecting LLM to Turing Machines and the application of the Rice Theorem as a scaffold to support the paper's main claims.

However, despite all such strengths, the paper also presents (imho) various weaknesses, which are discussed below.

**Weaknesses:**

## High-level
- It suffers from an "identity crisis" (it feels more like a "position" paper)
- Lack of a concrete experiment
- Some statements require further evidence to be supported
- The paper is built on a strong assumption that does not seem to have been accounted for
- The "mosaic prompts" are not really novel
- Some pieces of the text are unclear


## Comment

Despite my appreciation, I do have concerns about the suitability of this paper to ICLR'24. Before I discuss such concerns, however, I want to emphasize that my remarks are _my opinions_. I couldn't spot any technical or methodological flaw in the paper (which is also well-written): hence, my critiques are mostly directed at the "significance" aspect of the paper, and I endorse the authors to reflect on the following remarks. Ultimately, my goal is to help them make this paper as a noteworthy contribution to the state-of-the-art (be it for ICLR'24, or for any other venue).

### **Identity Crisis (Lack of a concrete experiment)**

The most prominent weakness is that, IMHO, the paper suffers from an "identity crisis" -- which is rooted on the fact that the paper touches both the "security" and "ML" domains.

On the "security" hand, all the considerations made in the paper are "obvious". The fact that, e.g., an attacker can bypass censorship mechanisms by inducing a LLM to output a "malicious set of actions" through individual prompts is "not new", and the fact that a similar strategy can fool essentially any precaution is "not surprising". Indeed, this is a well-known problem in reality, and the only way to solve this problem is by reading the attacker's minds. Plus, ultimately, LLM are just "tools": whether they are used in good- or bad-will is a different manner (and this had been known since the development of cryptographic protocols, since they also aid attackers in preventing their messages from being interpreted). So, to summarise, as a "security" researcher, the conclusion of this paper was already known, and the supporting theoretical arguments were hence somewhat redundant.

On the "ML" hand, the paper lacks a clear experiment that demonstrates at least one of the scenarios described in the ```practical implications```. Indeed, after introducing some definitions and demonstrating a given theorem, the paper merely limits to provide "thought experiments" discussing how an hypothetical attacker can achieve their goal. Yet, all such discussions are textual: there is an excessive usage of the words "can" "could" "may" "it is possible that". The paper does provide some references (e.g., "The authors of... showed that this can be done") but the lack of a concrete experiment is still hard to overlook. Such a lack is further aggravated by the additional what-ifs which project LLM into the future (e.g., ```these risks could become even more problematic```). I acknowledge that "anything can happen", but this is a weak argument.

Hence, I feel that the lack of a "hard" experiment is a significant weakness of this paper, which affects both its appeal to the security domain, as well as the one to the ML domain. For instance, I would have appreciated a clear demonstration of Figure 2 (I've spent ~30 minutes trying to have ChatGPT to process similar instructions, but I've never been successful).

Put differently, the paper currently reads as a "visionary paper" or a "position paper" rather than a true research paper. **However** do note that I am not saying that the paper is devoid of merit: providing "theoretical evidence" that it is not possible to craft "perfect" ML-based censorship mechanisms is a strong message.


### **Lack of evidence for some statements**

One of the major points in support of the "value" of this paper is that the current way to address censorship in LLM is by means of "ML-based mechanisms", and --after demonstrating that doing so will never guarantee 100% protection-- the suggestion that censorship should be treated as a security problem.

Indeed, to quote the abstract:

> Commonly employed censorship approaches treat the issue as a machine learning
problem and rely on another LM to detect undesirable content in LLM outputs.

The following was also stated in the Introduction:

> Such methods range from fine-tuning LLMs (OpenAI, 2023) to make them more aligned, to employing external censorship mechanisms to detect and filter impermissible inputs or outputs (Markov et al., 2023; Chockalingam and Varshney, 2023; Greshake et al., 2023).

However, I only see 4 works listed here. Hence, I wonder: is it really true that ML-based methods are the "way-to" address censorship problems? For instance, even Greshake et al. state ```Unfortunately, it is currently hard to imagine a foolproof solution for the adversarial prompting vulnerability```; moreover, the authors of NeMo Guardrails (used by NVIDIA (Chockalingam and Varshney, 2023)) state the following in their [GitHub repo](https://github.com/NVIDIA/NeMo-Guardrails/blob/main/docs/security/guidelines.md):

> Integrating external resources into LLMs can dramatically improve their capabilities and make them significantly more valuable to end users. However, any increase in expressive power comes with an increase in potential risk. To avoid potentially catastrophic risks, including unauthorized information disclosure all the way up to remote code execution, the interfaces that allow LLMs to access these external resources must be carefully and thoughtfully designed from a security-first perspective.

To me, the impression is that these mechanisms are proposed as a "partial" solution, since even the respective authors advocate for security principles to be followed. In light of this, the underlying "message" of the paper partially loses its value (at least imho). It would be enticing to carry out of more profound analysis of current works on approaches for LLM censorship, and pinpointing how many of such works truly claim to address censorship in an ML-only way, without making any security consideration: doing so would dramatically improve the contribution of this paper.


### **A strong assumption**

By looking at the definition of "censorship mechanism", the impression I have is that the paper assumes that censorship is always applied "a-posteriori". That is: the LLM receives an input, elaborates a response, and then --right before providing the response to the user-- it checks whether the response is permissible or not by means of some censorship. I wonder: is this really true?

Because, if this is not the case (i.e., there is some censorship applied to some "intermediate process" of the response), then the censorship would work, since it would be applied before the application of the transformation which makes the text encrypted.

In light of this, I invite the authors to provide evidence that this assumption holds _in reality_ (plus, I conjecture that such an observation CAN be used to develop some more effective defenses!). Otherwise, the authors should acknowledge that their analysis only applies to a specific use-case of censorship (do note that, however, this would decrease the impact of the paper). Alternatively, the authors can provide evidence (theoretical and, possibly, practical) that the envisioned analysis/findings hold even in these intermediate cases.

### **Naming of Mosaic prompts**

While I appreciate the name "Mosaic Prompts", I feel the way it is presented to be "excessive". Indeed, the described procedure is exactly the same as the "divide et impera" (or "divide and conquer") which is the de-facto praxis in computer science (and already associated to LLM, see [here](https://medium.com/@finomeno/exploring-large-language-models-insights-for-architects-393600dae131) and [here](https://medium.com/@digitalmiike/chatgpt-guide-10-effective-prompt-strategies-for-enhanced-output-979c8032eaaa)).

Hence, I endorse the authors to tone down this name, or at least acknowledge that it is just a renaming of a popular technique in computer science. (I am stating this also in light of the "acknowledgment" made in Footnote-1 -- which I greatly appreciated!)

### **Some pieces of text are unclear**

Although the paper is excellently written, I had issues in understanding some parts of the text. In what follows, I will directly quote each of these "problematic" parts, and explain the problems I encountered---starting from the Introduction.

> Such constraints can be semantic, e.g. does not provide instructions on how to perform illegal activities, or syntactic, e.g. does not contain any ethnic slurs from a provided set.

I did not understand the provided examples -- or rather, it is hard to determine the subject of the examples. I recommend rephrasing to, e.g., "the output must not provide..."

> methods against malicious attackers.

Are there attackers who are not malicious? (this redundancy occurs many times in the paper)

> restricting the string x to the set of permissible strings P

I recommend being more specific: "the string x to the set of permissible strings P that can be constructed by the LLM model" (otherwise, it may be confused with a string written by an user)

> demonstrated in Fig. 1

The caption states "Figure" (and not Fig.)

> typically defined by the language recognised it recognises

This is unclear

> descriptions of Turing machines can be viewed as a programming language, capable of being interpreted by a universal Turing machine capable of emulating them.

Please revise this statement as it is very confusing.

> As the semantic censorship impossibility result that we established by connecting the problem of semantic censorship to Rice’s Theorem doesn’t fully capture real world censorship settings where inputs and outputs are bounded we seek to provide another result on the impossibility of censorship that does.

Make this shorter, especially since the same message was written two lines before.

> we assert that given an invertible string transformation g

Is this "g" supposed to be the "bijective transformation"? Still, I am slightly confused about this "g" here; perhaps an example would be useful.

> it is capable of applying g to its output x to instead output g(x).

This is very unclear. Do you mean g(g(x))?

> either nothing is be permissible

Typo

> While existing LLMs are good at [...] Yuan et al. (2023)

This paragraph appers to be disconnected from the "Practical Implications". Or rather, it does not align well with the way the previous paragraph ended. Actually, I do not see any "practical implications" that are truly compellling here.

> While our results describe adversaries which can instruct

Which results?

> For example, users could provide [...] running the model

It would be wonderful if the authors showcased a way to do so in practice _today_.

> In an extreme setting where there exist only 2 permissible output strings

Why this assumption? To me, the following example holds even without this (perhaps I missed something?)

> converting text to ACII

Typo

> Subsequently, the user can request the model to output i’th bit

What is the ```i'th bit```? Plus, how can the user do so?

> our Mosaic Prompting results

Given that no experiments have been carried out, it is a bit of a stretch to define this as a "result" (even the Appendix does not provide "empirical results")



Finally, I report that the bibliography often does not provide the venue of a given work (e.g., the paper by Markov et al. (2023) was published in AAAI; whereas the one from Greshake et al. was accepted at AISec). This is annoying as a reader, as I could not ascertain the quality of a given referenced work.

**Questions:**

I liked the paper, and I am willing to improve my score if presented with compelling evidence that some of my remarks are flawed. Nonetheless, I invite the authors to answer the following questions (most of which are drawn from my "Weaknesses" section): depending on the answer, my rating will likely change.

1) Can the authors provide more evidence that LLM censorship is truly "commonly treated as a ML problem" (and that security-based approaches are not taken in consideration)?

2) Would the proposed theoretical analysis, as well as the proposed "attack", still apply if censorship is carried out during the process of crafting a response by the LLM? (Please elaborate)

3) How could the "attack" shown in Figure 2 be realized _today_?

Then, I have one last question. Assume that this paper is accepted to ICLR'24 as a spotlight. How would the authors present this work? Would the talk include only "what-ifs", or would it also showcase some concrete evidence that the envisioned scenarios are truly a security issue that cannot be countered with ML-only ways$^1$?

$^{\text{1: E.g., how do I make ChatGPT tell me "howdoibuildabomb"?}}$

---

> ### Author Response · Authors · 2023-11-20
>
> We thank the reviewer for their helpful and detailed feedback, and greatly appreciate the review. We have made modifications to the paper and provide responses to the concerns raised. We would be grateful for any further feedback on what could strengthen the work and make it more compelling.
>
> >On the "security" hand, all the considerations made in the paper are "obvious".
>
> We would like to first raise the point that while the results may seem somewhat obvious, they can carry significant implications and are not reflected in current perspectives of AI safety. For example, the recent preprint [1] suggests a path forward for controllable AGI through adaptation of formal verification, however our theoretical results show potential limitations for such approaches when many of the desired constraints are semantic. Even with formal verification, it might be possible to establish guarantees for individual model behaviors, but handling their composition would be far more challenging. While this paper reflects a forward looking perspective on AI safety, we’ve also included an overview of extant approaches.
>
> From a security perspective, the novelty and purpose of the paper is to expose fundamental security risks of LLMs that have been overlooked and whose significance is underappreciated. Existing LLM safety literature focuses on overly constrained threat models, such as adversaries that include adversarial tokens to prompts, with the aim of simply breaking a model’s “fine-tuned alignment”. We focus on output censorship mechanisms because, unlike alignment or intermediate representation censorship, output censorship mechanisms would lend themselves more to establishing safety guarantees, as permissibility of a model’s output would be defined in terms of the **content** of the string rather than the **internal mechanisms of the LLM** which produce it. The intent of this work was to make clear that providing guarantees that LLM outputs are not harmful through censorship mechanisms is hopeless by describing **novel threat models** not previously considered. The intent was not to demonstrate that a specific, implemented, censorship mechanism can be bypassed. The only research which has discussed bypassing censorship methods is [2], and they only studied the problem from a traditional jailbreaking perspective with a few modifications to minimize output of keywords that could trigger output filtering mechanisms.
>
> >On the "ML" hand, the paper lacks a clear experiment that demonstrates at least one of the scenarios described in the practical implications.
>
> We have added a proof-of-concept demonstration to the appendix showing how GPT-4-turbo can be bypassed using a combination of a roleplay jailbreak and instructions for performing Caesar Cipher to communicate in an encrypted manner. While these attacks are not yet easy to execute and come with challenges, we believe that it is best to be aware of these potential risks before they pose serious threats and thereby allow themselves to be rigorously, empirically, demonstrated. As such, we did not focus on developing benchmarks for these problems as such benchmarks could lead to a false sense of security from new censorship mechanisms which handle certain encryption methods, however, fail to provide guarantees as the underlying problem of censorship is inherently flawed.
>
> >It would be enticing to carry out of more profound analysis of current works on approaches for LLM censorship, and pinpointing how many of such works truly claim to address censorship in an ML-only way, without making any security consideration: doing so would dramatically improve the contribution of this paper.
>
> We’ve added a subsection to related works detailing current approaches to LLM defenses. While the approach of censorship has not been formalized prior to this work, many extant defense methods are implicitly defining an instance of an input or output censorship mechanism which relies on an LLM to evaluate harmfulness of strings. There is of course awareness that these methods are imperfect, however there is no awareness that such defense mechanisms are fundamentally limited as we describe. If the reviewer could describe more specifically what they mean by a more profound analysis of current methods, we would be happy to expand further on this point.

---

> ### Author Response · Authors · 2023-11-20
>
> >Hence, I endorse the authors to tone down this name, or at least acknowledge that it is just a renaming of a popular technique in computer science.
>
> We acknowledge that Mosaic Prompts are related to decomposed prompting strategies that have been studied for improved problem solving, however the formulation is more general than question decomposition, has a different motivation for the decomposition, and stresses decomposition across context windows. We have added related work outlining this.
> We are open to renaming Mosaic Prompts to Dual-use or Dual-intent Decomposed Prompting if this would still be preferred, the latter was used by the Code LLama [6] paper when referring to attacks described in this work *(we ask the reviewers to not check the reference to so as to not deanonymize the authors)*.
>
> Questions:
> 1. >Can the authors provide more evidence that LLM censorship is truly "commonly treated as a ML problem" (and that security-based approaches are not taken in consideration)?
>
> See updated related works section
>
> 2. >Would the proposed theoretical analysis, as well as the proposed "attack", still apply if censorship is carried out during the process of crafting a response by the LLM? (Please elaborate)
>
> Censorship mechanisms that aim to intervene on intermediate states of an LLM, e.g. Representation Engineering [3] approaches such as [4] would not be subject to the theoretical limitations. Generally, such methods may pose a practical way forward for further improving security but become subject to similar limitations as model alignment. In particular, one cannot provide guarantees of safety as permissibility of an output would not be determined by hidden representations of an LLM but rather by the string itself, i.e. we assume there is a ground truth for permissibility. Furthermore, the vulnerability of such methods to adversarial prompts, which already pose significant challenges to model alignment, is unclear and would require further study and analysis. In other words, just as one would not define a cat in terms of the internal representations of a classifier, we cannot define impermissible content through internal representations, such definitions would lead to counter-examples via adversarial perturbations.
> Once LLMs are effectively integrated with some form of memory tape or scratchpad where they can store information, the ability to monitor and audit that scratchpad may also provide options for defense, however it is difficult to draw conclusions without knowing how these mechanisms will work and be used.
> We note that the aforementioned potential avenues for defense would not affect Mosaic Prompting attacks.
>
> 3. >How could the "attack" shown in Figure 2 be realized today?
>
> We’ve included an example of how the attack shown in Figure 2 can be realized by replicating a version of the attack demonstrated [5] alongside the addition of a simple handcrafted roleplay jailbreak aimed to break model alignment as the concern of our work is on model censorship and attacks need to be combined with Jailbreaking methods to bypass model alignment measures.
>
> 4. >Assume that this paper is accepted to ICLR'24 as a spotlight. How would the authors present this work? Would the talk include only "what-ifs", or would it also showcase some concrete evidence that the envisioned scenarios are truly a security issue that cannot be countered with ML-only ways
>
> While the aim of this work was to present fundamental limitations of LLM safety to encourage a forward looking re-evaluation of the problem, design threat modeling, and defenses, we can present a preliminary proof-of-concept example as added to the appendix to demonstrate that one can already abuse these limitations. We see our work as a foundational step towards defining strongest adversaries that our systems should ultimately attempt to address to stop manipulation and abuse.
>
> [1] Tegmark, Max, and Steve Omohundro. "Provably safe systems: the only path to controllable AGI." arXiv preprint arXiv:2309.01933 (2023).
>
> [2] Deng, Gelei, et al. "Jailbreaker: Automated jailbreak across multiple large language model chatbots." arXiv preprint arXiv:2307.08715 (2023).
>
> [3] Zou, Andy, et al. "Representation engineering: A top-down approach to ai transparency." arXiv preprint arXiv:2310.01405 (2023).
>
> [4] Belrose, Nora, et al. "Eliciting latent predictions from transformers with the tuned lens." arXiv preprint arXiv:2303.08112 (2023).
>
> [5] Yuan, Youliang, et al. "Gpt-4 is too smart to be safe: Stealthy chat with llms via cipher." arXiv preprint arXiv:2308.06463 (2023).
>
> [6] Roziere, Baptiste, et al. "Code llama: Open foundation models for code." arXiv preprint arXiv:2308.12950 (2023).

---

> > ### Comment · Reviewer_BZEx · 2023-11-21
> > **Ack**
> >
> > I thank the authors for their response and for updating the paper. I have briefly gone through the revised text, and I am pleased by the added experiment which does provide evidence that the envisioned scenario is "practically realizable" (at least to some extent).
> >
> > Nonetheless, I found that the revised version does not make any mention of the answer to my second question (correct me if I'm wrong). I feel that this is an important point that should be accounted for in the paper -- both as a limitation, but also as an avenue for future work. Hence, I endorse the authors to improve the current manuscript by elaborating on this aspect.
> >
> > Furthermore, about "Mosaic Prompts", my suggestion was merely to acknowledge that the technique borrows from existing work. I think this is out of fairness (the paper's contributions would not be diminished in any way with this).
> >
> > Finally, there are some minor typos/formatting issues (e.g., ```cite...``` in the Related Work; and there is also a trailing space after each instance of colored text---likely due to some LaTeX macro).

---

> > > ### Author Response · Authors · 2023-11-21
> > >
> > > Thank you very much for the pointers. We have updated the submission to address the reason why we do not look at intermediate censorship in the introduction. We also address this question in more detail in our response.
> > >
> > > We have also corrected the typo's, and thank you for spotting them.

---

> > > > ### Comment · Reviewer_BZEx · 2023-11-23
> > > > **Ack**
> > > >
> > > > Thank you for the improvement. I will take them into account when discussing the paper with the other reviewers and AC.

---

### Official Review · Reviewer_ST3b · 2023-10-25

**Soundness:** 3 good
**Presentation:** 2 fair
**Contribution:** 2 fair
**Rating:** 5
**Confidence:** 2

**Summary:**

This paper investigates the theoretical limitations of the current external censorship mechanisms in LLMs from the view of computing theory. Given these inherent limitations, the authors argue that LLM censorship should be addressed more as a security problem than a machine learning problem.

**Strengths:**

- Trendy topic
- A novel perspective to study LLM censorship

**Weaknesses:**

- Implications can be extended
- Readability can be improved

**Questions:**

In this paper, the authors first focus on the semantic censorship mechanisms, proving that the current mechanisms cannot reliably detect if LLM output is "semantically impermissible." They further show that such limitations are inherent and can extend beyond semantic censorship mechanisms by designing Mosaic prompts.

Overall, the authors study a trendy topic and offer a novel perspective to understand LLM censorship. However, I have the following concerns.

- The authors prove the impossibility of semantic censorship using string transformation by showing how the transformed string might break the "invariance of semantic censorship." Here, I have some doubts regarding the invariance property. In my opinion, the semantics of a string often change after the transformation. Thus, it is reasonable for the transformed string to bypass semantic censorship mechanisms. Moreover, LLMs do not necessarily output harmful texts with the transformed string. Why does the invariance property hold? Is this property an important goal considered by LLM censorship developers when designing their mechanisms?

- Implications can be extended. It appears to me that the current implication discussion stops at showing LLM censorship is more of a security problem than a machine learning problem. What are the direct implications for model developers when building censorship? Are there any defensive measures against the Mosaic prompts? The authors only briefly mention that there are standard approaches, such as access controls and user monitoring, to build censorship from the security view. However, there is no further analysis showing that these approaches can indeed overcome the theoretical limitations of current external censorship mechanisms and surpass them in censorship performances.

- Readability can be improved. Many sentences are too long and difficult to read. For example, "Thus, we can understand censorship as a method of determining permissibility of a string and censorship mechanisms can be described as a function, f(x), restricting the string x to the set of permissible strings P by transforming it to another string x' ∈ P if necessary, e.g. x' ='I am unable to answer.'"

---

> ### Author Response · Authors · 2023-11-20
>
> We appreciate the reviewers' insightful questions and helpful feedback for improving the work, we hope that our updated draft and responses help address the concerns.
>
> >The authors prove the impossibility of semantic censorship using string transformation by showing how the transformed string might break the "invariance of semantic censorship." Here, I have some doubts regarding the invariance property. In my opinion, the semantics of a string often change after the transformation.
>
> We acknowledge the point made regarding the semantics of an output string changing upon undergoing a string transformation, which is why we specify the invariance holds under knowledge of $g^{-1}$. What this means is that harmful text may be recoverable from apparently benign text if a specific transformation is known. We’ve provided an example in the appendix demonstrating how one can communicate with GPT-4-turbo using a Caesar cipher, and thereby recover an impermissible output from encrypted text that on it’s own is semantically nonsensical and perceived as harmless when evaluated by an independent LLM. The central theme of our work is demonstrating that harmlessness and safety cannot be evaluated independent of knowledge of a broader context. From seemingly benign output strings that could be transformed into harmful content under invertible transformations to benign outputs which are part of a collection that in aggregate leads to construction of impermissible content. As we show, one can extract dangerous information from LLMs through these methods, which necessitates a re-evaluation for how AI safety is studied.
>
> >Implications can be extended.
>
> See the updated appendix for a more extensive discussion and explanation on the implications for developers building censorship mechanisms. The intent of such approaches isn’t to overcome the theoretical limitations of current censorship mechanisms, rather, they are intended to allow for the provision of certain types of output guarantees, clarifying what statements regarding the safety and vulnerability of a deployed LLM can and cannot be made.
> Censorship mechanisms against Mosaic Prompt are infeasible, however, prompt templates result in strict constraints on the space of possible outputs and information accessible by users.
>
> >Readability can be improved.
>
> Thank you for the feedback regarding readability, we’ve updated the draft.

---

### Official Review · Reviewer_3fNc · 2023-10-29

**Soundness:** 2 fair
**Presentation:** 2 fair
**Contribution:** 1 poor
**Rating:** 3
**Confidence:** 3

**Summary:**

The paper's topic studying censorship and its effectiveness is interesting, ie. what kinds of knowledge can be extracted from LLMs and whether protection mechanisms can be circumvented. But the paper contributes little of practical value. It also lacks a proper evaluation to claims and conceptual illustrations. The theoretical treatment would be interesting, but the paper claims are mostly direct implications of existing theorems or require minor enhancements. Overall, the contribution appears marginal.

Details:
* abstract:  LM -> LLM or define it.
*  The example, Figure 1 is not of any practical value and might be conceptually it is flawed - the three steps are the least challenge in making successful ransomware attack (deploying it is much more of an issue, avoiding being detected too). The Mosaic prompt is also not very convincing. Both should be shown to be actually working.
* The idea to use encryption (Appendix A) is interesting, but is this a practical concern? Does it add to the discussion of how protection mechanisms can be circumvented? It might, if it was shown to work. But as is, it seems incomplete.
* On a high level, the paper argues that censorship cannot work because a malicious person might not directly asked for censored actions, but for steps needed for these actions, which might not be censored. But this holds for almost anything in our world and is nothing new. Any technological knowledge can be abused.  A knife can be used to kill or to save a life (doctor during surgery).  A motor can power an ambulance saving life or a truck performing a terrorist act. This is general knowledge. The paper seems to sell this as a novel aspect. The fundamental question is: Should knowledge and technology be made available that can be abused?  This is also not really a security question as the paper argues. Obviously any abuse relates to security, but I don't see, why the paper's claim to say "LLM censorship (ie. avoiding censorship through attacks) is a security concern" should be a new insight.

**Strengths:**

see above

**Weaknesses:**

see above

**Questions:**

see above

---

> ### Author Response · Authors · 2023-11-20
>
> We thank the reviewer for the feedback provided.
>
> >It also lacks a proper evaluation of claims and conceptual illustrations. The theoretical treatment would be interesting, but the paper claims are mostly direct implications of existing theorems or require minor enhancements.
>
> We’ve provided a demonstration of how encryption could be used to produce encrypted outputs that can pass censorship detectors to the appendix. While theoretical results follow directly from prior results or the definitions and problem formulation, we do not believe this should be viewed as a limitation and instead can provide clarity and understanding of the very fundamental issues of LLM safety and security. We would appreciate any suggestions regarding which theoretical results would make our work more convincing.
>
> >the three steps are the least challenge in making a successful ransomware attack (deploying it is much more of an issue, avoiding being detected too).
>
> While ransomware deployment is certainly a challenge, the example in Figure 1 was intended to provide an illustration of how one can acquire information for the creation of ransomware, information that is normally impermissible, from an LLM through permissible questions and outputs. Our intent was not to demonstrate that extant LLMs can be used directly for conducting harm, merely to illustrate the challenge of guaranteed control of LLM usage which has been highly sought after and actively researched in recent years.
>
> We do want to note however, that aside from obvious complexity in running ransomware, developing correct ransomware took a bit of time for the criminals. Doing cryptography correctly is hard, even for very simple old settings [1,2]. A number of ransomware operators made mistakes in how they handled command and control, key selection and generally hardcoded secrets [3,4]. It is not unreasonable to imagine that code assistants will help fix some of the simple mistakes or generally reduce the cost and barriers to creating ransomware.
>
> >The idea to use encryption (Appendix A) is interesting, but is this a practical concern? Does it add to the discussion of how protection mechanisms can be circumvented?	 It might, if it was shown to work
>
> The encryption with Diffie-Hellman exchange does not appear to be an imminent concern for extant LLMs available due to limitations of LLMs to perform complex tasks such as handling storage of values not explicitly written down and performing modular arithmetic error free, however we believe it could be of interest for future models several years down the line or could be outsourced to a separate tool, with intriguing implications arising from the ability to establish encrypted communication with an LLM with strong cryptographic guarantees.
>
> >On a high level, the paper argues that censorship cannot work because a malicious person might not directly asked for censored actions, but for steps needed for these actions, which might not be censored. But this holds for almost anything in our world and is nothing new. Any technological knowledge can be abused
>
> When making a technology or knowledge available, it is prudent to understand how it could be abused, the potential threats, and to make use of any methods that can make the technology safer. The aim of this paper, as with many works in LLM Safety literature, is to provide a better understanding of the potential for misuse and abuse of current and future LLMs, as well as an understanding of how LLMs can (or more importantly cannot) be made safe or secure . We believe our work provides a significant contribution to the area by articulating approaches that malicious users could utilize to bypass censorship mechanisms regardless of the type of mechanism employed. We also provide a formalized description of LLM defenses via censorship, and in doing so enables us to clearly show why the problem is inherently ill-posed.
>
> [1] Whitten, Alma, and J. Doug Tygar. "Why Johnny Can't Encrypt: A Usability Evaluation of PGP 5.0." USENIX security symposium. Vol. 348. 1999.
>
> [2] Sheng, Steve, et al. "Why johnny still can’t encrypt: evaluating the usability of email encryption software." Symposium on usable privacy and security. ACM, 2006.
>
> [3] Tilly Travers. “Ransomware mishaps: adversaries have their off days too” (https://news.sophos.com/en-us/2021/08/11/ransomware-mishaps-adversaries-have-their-off-days-too/)
>
> [4] Jessica Lyons Hardcastle. “Good news for Key Group ransomware victims: Free decryptor out now” (https://www.theregister.com/2023/08/31/key_group_ransomware_decryptor/)

---

> ### Comment · Reviewer_3fNc · 2023-11-21
> **Acknowledgement**
>
> Thank you for the reply. I acknowledge the responses and I have read the other reviews and responses. Overall, given mostly the relevance of the topic, I would be willing to upgrade the rating from 3 to 4, which is unfortunately not existing.

---

### Official Review · Reviewer_xHLz · 2023-10-29

**Soundness:** 2 fair
**Presentation:** 4 excellent
**Contribution:** 3 good
**Rating:** 5
**Confidence:** 3

**Summary:**

This paper explores some of the theoretical limitations of LLM censorship, the problem of identifying permissible inputs and outputs to language models. In particular, the paper focuses on the limitations of semantic censorship, or filtering of strings based on their meaning. First, the paper shows that determining whether a “program” output by an LLM is permissible is an undecidable problem. Then, the paper discusses the impossibility of semantic censorship by showing that strings can undergo transformations which preserve their semantic meaning but are otherwise unintelligible except to a user who knows how to invert the transformation. Finally, the paper introduces Mosaic Prompts, a way of breaking up an impermissible prompt into permissible pieces.

**Strengths:**

This paper’s primary strength is that it identifies an important issue to focus on that has been unexplored in the literature - what are the theoretical limits on the ability to filter LLM inputs or outputs based on their semantic meaning? The paper is a good exposition of this problem and the theoretical settings it considers highlight some important limitations for the task. The figures and tables also do a good job of clarifying some of the concepts in the text. Overall, the authors’ assertion that syntactic censorship is likely to be more successful than semantic censorship is well-taken from this work.

**Weaknesses:**

This paper’s primary weakness is the number of assumptions and limitations that come into the different theoretical treatments that the paper covers. First, the paper itself admits that the treatment of Rice’s theorem for programs on Turing Machines is not generally applicable to the bounded inputs and outputs case of LLMs. Second, in the section 2.2 on the invertible transform, I believe there may be a flaw in the reasoning of the proof. Under assumption 1, the authors assume that the model is capable of following instructions such that it can produce the transformation $g$. This assumption is explicitly stated. It seems that the proof also requires that the LLM (or corresponding companion LLM that is doing censorship) is unable to compute the inverse transformation $g^{-1}$. If it were, then it could check the semantics of the un-transformed string for permissibility. This assumption weakens the power of the impossibility result in my opinion. Finally, while I think that the Mosaic Prompt approach is interesting, I do think the paper underestimates the LLM’s ability to attend to previous prompts. While in the mosaic approach the model is likely to answer early prompts, it is conceivable that once enough of the pieces of the impermissible prompt are present, one would be able to detect the impermissibility of the conversation overall.

**Questions:**

Does the impossibility result in Section 2.2 require an assumption that $g^-1$ is not computable by the permissibility model?

Is the problem space simplified at all by considering the compositionality of strings? For example, if there is an impermissible substring within a larger string, does that make the larger string automatically impermissible as well?

Does something like “fuzzy” permissibility fit into this framework at all? For example, many prompts and outputs would be considered “borderline” or have some level of “toxicity” if sent to a human rater, rather than a bright-line permissible vs. not rule. Does that make the problem any easier or harder?

---

> ### Author Response · Authors · 2023-11-20
>
> Thank you for your thoughtful review and thoughtful questions and feedback. We hope that our response and submission revisions address your questions.
>
> >Does the impossibility result in Section 2.2 require an assumption that $g^{-1}$ is not computable by the permissibility model?
>
> It is perfectly fine that the model is capable of computing the inverse transformation $g^{-1}$. The fact that $g^{-1}$ of the output is impermissible was already established as the model internally crafted that content before providing the encrypted output. The impossibility comes from the fact that given only the output, one does not know how it came about; in other words, there are infinitely many different schemes that could have given rise to a given output, only some of which are impermissible. The issue stems from a lack of knowledge rather than lacking computational capability to invert the transformation.
>
> >I do think the paper underestimates the LLM’s ability to attend to previous prompts. While in the mosaic approach the model is likely to answer early prompts, it is conceivable that once enough of the pieces of the impermissible prompt are present, one would be able to detect the impermissibility of the conversation overall.
>
> The power of the mosaic prompting approach stems from the ability of a user to decompose questions into separate context windows, which makes it impossible to keep track of which user interactions are connected or not, as model outputs can later be composed/aggregated by the user to attain impermissible content. A user could even create many different accounts to ask a question from each one, thus even with perfect individual user monitoring it would be impossible to know if a given individual is acquiring pieces of information which contribute to the acquisition of dangerous information. In the field of computer security it is common to assume that it is cheap/easy to create as many users/identities as needed, something referred to as sybil attacks.
>
>
> >Is the problem space simplified at all by considering the compositionality of strings? For example, if there is an impermissible substring within a larger string, does that make the larger string automatically impermissible as well?
>
> The problem space cannot be simplified by considering the compositionality of strings since the Mosaic Prompting issue leverages the ability to compose information rather than just extracting impermissible substrings from output strings. As we described in Section 3.3, even if a model can only produce two permissible outputs it could still be possible to extract impermissible content bit by bit. The ability to extract impermissible content from a permissible string, e.g. extracting an inappropriate substring from a permissible string such as “assume” could also be seen as an issue, however the larger issue is the ability for a user to attain content that was never directly provided by the network. We highlight that LLMs should be seen as a tool for harm and security analyses should be viewed from this perspective rather than just a question of whether or not the LLM produces harmful and dangerous responses directly.
>
> >Does something like “fuzzy” permissibility fit into this framework at all?
>
> Fuzzy permissibility would not solve these issues simply because at the end of the day one has to determine what output can or cannot be returned to the user unfiltered; this imposes the notion of permissible and impermissible content that we work with.

---

### Author Response · Authors · 2023-11-20

We're very grateful to the reviewers for their feedback, comments, and questions. We have updated the submission, text in red corresponds to fixed content, orange is shortened content, and blue is newly added content. We are happy to answer any further questions and would be grateful for any further feedback regarding the submission and how it can be improved.

---

### Meta-Review · Area_Chair_Z2Zq · 2023-12-03

**Metareview:**

The paper studies the problem of LLM censorship, which is to use an algorithm, potentially implemented by another LLM, to determine whether a string is permissible. The paper provides an interesting theoretical argument: LLM censorship is actually an “undecidable” problem, which means that no algorithm exists to fulfill this goal.
The paper then presents a “Mosaic Prompts” method to bypass the LLM censorship, which decomposes an impermissible program into permissible components.
Finally, the paper argues that LLM censorship should be dealt with by security approaches instead of ML approaches.

**Justification For Why Not Higher Score:**

The main reason that the reviewers reject this paper is due to limited contributions. The theorems are mostly from direct implications of existing ones. The paper also lacks sufficient evaluations to support the proposed Mosaic Prompts approach, which by far seems more like a proof of concept. While the paper's ideas are generally very interesting, as pointed out by one reviewer, it feels more like a "position" paper instead of a scientific contribution suitable for ICLR.

**Justification For Why Not Lower Score:**

N/A

---

### Decision · Program_Chairs · 2024-01-16

Reject